# Free Radical Scavenging Activity of Infusions of Different Medicinal Plants for Use in Obstetrics

**DOI:** 10.3390/plants10102016

**Published:** 2021-09-26

**Authors:** Sylwia Jarco, Barbara Pilawa, Paweł Ramos

**Affiliations:** 1Department of Biophysics, Faculty of Pharmaceutical Sciences in Sosnowiec, Medical University of Silesia, Katowice, Jedności 8, 41-200 Sosnowiec, Poland; sylwia.jarco@med.sum.edu.pl (S.J.); pawelramos@sum.edu.pl (P.R.); 2Non-Public Health Care Mamma sp. j. Sylwia Jarco, Lotnicza 19, 43-300 Bielsko-Biała, Poland

**Keywords:** free radicals, antioxidant, infusion of medicinal herbs for obstetrics, *Asparagus racemosus*, *Mitchella repens*, *Cnicus benedictus* L., *Galega officinalis* L., *Eupatorium cannabinum* L., UVA radiation, EPR spectroscopy

## Abstract

An X-band (9.3 GHz) electron paramagnetic resonance (EPR) spectroscopy was used to examine the free radical scavenging activity of the following infusions, which were nonirradiated and exposed to UVA: root of *Asparagus racemosus* and herbs of *Mitchella repens*, *Cnicus benedictus* L., *Galega officinalis* L., and *Eupatorium cannabinum* L. The plant materials for obstetrics applications were chosen for analysis. The aims of these studies were to compare the free radical scavenging ability of the tested infusions and to determine the influence of UVA irradiation of the plant materials on interactions of these infusions with free radicals. Both the magnitude and kinetics of the interactions of the infusions with the model DPPH free radicals were examined. The ability to quench the free radicals for the examined plant infusions increases in the following order: *Asparagus racemosus* (root) < *Mitchella repens* (herb) < *Cnicus benedictus* L. (herb) < *Galega officinalis* L. (herb) < *Eupatorium cannabinum* L. (herb). The analyzed infusions differ in the kinetics of the interactions with free radicals. The fastest interactions with free radicals characterize the infusions of *Galega officinalis* L. herb and *Eupatorium cannabinum* L. herb. The infusion of *Mitchella repens* herb interacts with free radicals in the slowest way. UVA radiation reduces the antioxidant interactions of all tested infusions, especially the infusion of *Eupatorium cannabinum* L. herb, which should be protected against UVA radiation during storage. The weakest decrease of free radical scavenging activity was observed for the infusion of the root of *Asparagus racemosus* exposed to UVA radiation. UVA radiation affected the speed of the free radical interactions of the infusions, depending on the type of plant materials. EPR spectroscopy is useful to examine the free radical scavenging activity of plant infusions, which is helpful to find effective antioxidants for applications in obstetrics and their optimal storage conditions.

## 1. Introduction

Medicinal plants, due to the content of various chemical compounds, affect the human body and one of the types of their influence is antioxidative interaction [1,2,3,4,5,6,7,8]. Antioxidants include flavonoids, phenolic acids, anthocyanins, procyanides, isoflavin, stilbenes, tocopherols, carotenoids (astaxanthin, lycopene, lutein, β-carotene), sarotenoids, tannins, coumarins, vitamins A, C, and E, and bioelements: selenium, zinc, copper, and manganese [1,2,3,4,5,6,7,8,9]. In this work, interactions with free radicals were examined for infusions of the plant materials that are important from the point of view of obstetrics: *Asparagus racemosus* (the root), *Mitchella repens* (herb), *Cnicus benedictus* L. (herb), *Galega officinalis* L. (herb), and *Eupatorium cannabinum* L. (herb).

The aims of this work were to compare free radical scavenging activity of the tested plant infusions and to determine the influence of UVA irradiation of the plant materials on the antioxidant properties of the infusions. The examined plants are used during the lactation process and to relieve female ailments [10,11,12,13,14,15]. Examination of their antioxidant properties is justified, because of the increase in energy metabolism during pregnancy, and lactation promotes the formation of free radicals in the female body [10,11,12,13,14,15]. A woman’s food has antioxidant properties, and the concentration of antioxidants in breast milk depends on the consumed foods and drugs [13,16,17,18,19]. Studies of the interactions of the selected plant infusions with free radicals were performed to evaluate their usefulness as potential antioxidants for obstetrics.

*Asparagus racemosus* contains a lot of organic compounds: steroid saponins, glycosides, alkaloids, polysaccharides, mucus, racemosol, isoflavones, and small amounts of minerals: zinc, manganese, copper, cobalt, and selenium [9,20]. *Asparagus racemosus* can be found in Indian Pharmakopeia and Great Britain Pharmakopeia [21]. In medicine, powder, infusion, and brew of the root of *Asparagus racemosus* are used [20]. They are applicable for women in pregnancy, puerperium, and during lactation [9,20]. The root of *Asparagus racemosus* is used in women with inflammation of the genital organs, in hormonal disorders, premenstrual syndrome, polycystic ovary syndrome and in infertility, at risk of miscarriage, and in menopause [20,21]. The root of *Asparagus racemosus* is recommended in case of weakened lactation [20,22,23,24]. The root of *Asparagus racemosus* is used in inflammatory states of mucosal, lungs, stomachs, kidneys, in diarrhea, diabetes, hepatopathy, and nervous disorders [20,21,25]. Extracts from the root of *Asparagus racemosus* have antioxidant properties [23,26,27].

*Mitchella repens* contains resin, wax, mucus, dextrin, saponins, alkaloids, glycosides, and tannins [23,28]. *Mitchella repens* herb was already used by Indians in North America to prevent miscarriage, pregnancy complications, and to reduce pain in labor [23]. *Mitchella repens* herb has sedative [23] and diuretic [29] effects. Extracts of *Mitchella repens* herb are typically used in the puerperium to alleviate nipple swelling and pain, and in the burn of the breasts [23,29]. It was stated in [23] that there are no literature reports about safety of use of *Mitchella repens* herb in pregnant women and during lactation.

*Cnicus benedictus* L. contains flavonoids, polyacetylates, lactones, essential oil, and mineral compounds, mainly calcium and magnesium salts [9]. *Cnicus benedictus* L. herb is used as an aid to increase lactation [23,24,30], as a mean of accelerating menstruation [22], and it prevents hemorrhages in childbirth [31]. *Cnicus benedictus* L. herb is used in digestive tract diseases [9,30,31,32], in the case of deficiency of digestive juices [9,32,33], indigestion [32,33], biliary tract and liver disorders [9], and in disorders of the pancreas [32]. *Cnicus benedictus* L. herb has antibacterial [9,22], antiviral [9], diuretic [9], and antipyretic properties [33]. These herbs are recommended in the case of anemia and deficiency of minerals [9]. *Cnicus benedictus* L. herbs should not be used during pregnancy [33]. Side effects may include nausea, vomiting, and diarrhea [9,22]. Interactions with insulin, proton pump inhibitors, and ulcer drugs are possible [22].

*Galega officinalis* L. contains flavones glycosides, guanidine derivatives, chromium salts, tannins, and phytoestrogens: kaempherol and quercetin [9,30,34]. *Galega officinalis* L. herb is used to stimulate lactation in women [30,35,36,37], because it increases the secretion of prolactin [30] and improves blood circulation in the mammary gland [35]. The water extract from the fresh plant significantly increases milk production [36]. *Galega officinalis* L. herb decreases the blood sugar levels [9,33,34], regulates the adrenal cortex [9], and it works as a diuretic [22,34]. Guanidine and peganine derivatives were found in some of the ground part of the plant [28]. The known side effects of the use of this herb are headaches and emotional tension [22].

*Eupatorium cannabinum* L. contains flavonoids (eupatorine), pyrrolizidine alkaloids (supinin, lycopsamine), lactones, chlorogenic acid, and heteroxylans [33,38]. This herb is used in women with inflammation of the uterus and fallopian tubes [29]. Infusions, brews, and inhalations with *Eupatorium cannabinum* L. herb are indicated in case of fever and cold [33]. The pyrrolizidine alkaloids can cause a risk of carcinogenity and hepatotoxicity [33]. However, *Eupatorium cannabinum* L. herb is recommended for diseases of the liver and gall bladder [33].

Particular attention has been paid to UV radiation, because it affects the stability of drugs [39,40], and it causes the largest loss of the active structure of melatonin as a drug [40]. UVA radiation increases the risk of skin cancer [41]. In this work, it was assumed that the interactions of antioxidants with free radicals not only depend on their type [42,43,44,45], but also the physical factors in the environment, including UV radiation, which affect these interactions [46,47]. We searched for answers to the question of whether the tested plant raw materials can be stored with access to UVA radiation. The assumption of the research was that the conditions of storage, including UVA radiation, should not change the antioxidant interactions of the medicinal infusions. The effects of UVA radiation of the tested plant raw materials on the free radical scavenging activity of their infusions are not known so far. The direct method of testing free radicals has been proposed by us. We chose the physical method for research: the electron paramagnetic resonance (EPR) spectroscopy, which is not destructive for the samples [48,49,50]. The interactions of the antioxidant species with free radicals decrease microwave absorption, which is reflected in changes of the EPR spectra.

## 2. Results and Discussion

The performed EPR examination showed that the tested plant infusions behave in contact with DPPH free radicals as antioxidant species. All the examined infusions of the plant materials, which were not exposed to UV radiation, quench the EPR signal of DPPH free radicals. The magnitude of this effect depends on the type of the raw material. The amplitude of the EPR spectra of free radicals depends on the number of free radicals in the samples [48,49,50]. The quenching of the EPR spectra by antioxidants results from the lower contents of free radicals in the object located in the resonance cavity of the EPR spectrometer. The EPR spectra are measured, because unpaired electrons of free radicals in the magnetic field absorb microwave energy, they excite to the higher energy levels, and then the electrons go to lower levels as a result of magnetic relaxation processes [48,49,50].

For example, the EPR spectra of DPPH free radicals interacting with the infusion of the root of *Asparagus racemosus,* which was not exposed to UV radiation for interaction times 3, 9, 15, 21, 27, and 33 min, are presented in Figure 1. The g-values were equal to 2.0036. The EPR spectra of DPPH free radicals changed with increasing interaction time (*t*) for the infusion of the root of *Asparagus racemosus* that was not exposed to UV radiation. As one can see, the EPR spectra of DPPH free radicals decreased with the increase of their contact with the infusion of the root of *Asparagus racemosus* and then the spectra stabilized and they did not change with a further increase of time. The time evolution of the EPR spectra of DPPH in Figure 1 indicates that the infusion of the root of *Asparagus racemosus* initially quenched the DPPH free radicals increasingly more, and finally the quenching stabilized, the EPR spectra became the lowest, and the EPR spectra of free radicals remained constant in time. Similar changes of the EPR spectra of DPPH free radicals in contact with infusions of the nonirradiated herbs: *Mitchella repens*, *Cnicus benedictus* L., *Galega officinalis* L., and *Eupatorium cannabinum* L., were observed.

The kinetics of the free radical interactions changes with the relative amplitudes (A/A_DPPH_) of the EPR spectra of DPPH free radicals with increasing time (*t*) of interaction with the infusions of the root of *Asparagus racemosus*, *Mitchella repens* herb, *Cnicus benedictus* L. herb, *Galega officinalis* L. herb, and *Eupatorium cannabinum* L. herb for the plant materials not exposed to UV radiation are compared in Figure 2. All values of the spectral amplitudes (A) measured for DPPH free radicals in contact with the infusions were divided by the values of the amplitude (A_DPPH_) of the EPR spectrum of DPPH free radicals in the reference solutions. The relative amplitudes (A/A_DPPH_) of the EPR lines of DPPH free radicals interacting with the plant infusions are lower values than 1 (Figure 2). The values of (A/A_DPPH_) lower than 1 indicate quenching of DPPH free radicals, so also their EPR signals, by the antioxidative plant infusions.

In the first phase of the interactions between DPPH free radicals and the plant infusions, the relative amplitudes (A/A_DPPH_) decrease increasingly more, because an increasing number of free radicals are being quenched. In the second phase of kinetics in Figure 2, the relative amplitudes (A/A_DPPH_) reach the minimum and their values remain constant. The stronger antioxidants have lower minimal values of relative amplitudes (A/A_DPPH_). Looking at the results in Figure 2, one can conclude that the lowest minimal relative amplitude (A/A_DPPH_) has an EPR line of DPPH free radicals in contact with the infusion of *Eupatorium cannabinum* L. herb, and this infusion shows the strongest interaction with free radicals. The highest minimal relative amplitude (A/A_DPPH_) has an EPR line of DPPH free radicals in contact with the infusion of the root of *Asparagus racemosus*, so this infusion shows the weakest quenching of DPPH free radicals.

The values of the minimal relative amplitudes (A/ADPPH) of the EPR spectra of DPPH free radicals interacting with the infusions of the root of *Asparagus racemosus*, *Mitchella re-pens* herb, *Cnicus benedictus* L. herb, *Galega officinalis* L. herb, and *Eupatorium cannabinum* L. herb not exposed to UV radiation are compared in Figure 3. For the interactions of free radicals with the infusions of the nonirradiated plant materials, the value of the minimal relative amplitude (A/ADPPH) of the EPR spectra of DPPH free radicals decreases in the following order: infusion of the root of *Asparagus racemosus* > infusion of *Mitchella repens* herb > infusion of *Cnicus benedictus* L. herb > infusion of *Galega officinalis* L. herb > infusion of *Eupatorium cannabinum* L. herb. So, interactions of the examined infusions with free radicals increase in this order. Considering the plant materials not exposed to UV radiation, interactions with free radicals are the weakest for the infusion of the root of *Asparagus race-mosus*, and they are the strongest for the infusion of *Eupatorium cannabinum* L. herb (Figure 3). The EPR spectrum of DPPH free radicals in contact with the root of *Asparagus racemosus* has a considerably higher relative minimal amplitude (A/ADPPH) in relation to the other lines, and it is a weaker antioxidant. The infusion of *Eupatorium cannabinum* L. herb has the best antioxidant properties, which follow from the lowest value of the relative minimal amplitude (A/ADPPH). It should be emphasized here that this work present results from the physical point of view, and the application of antioxidants to obstetrics should be developed by specialists of other areas. The studies were performed in vitro and the results have limitations related to the in vivo situation. For example, the antioxidant compounds of the plant infusions may display an altered stability in tissues. The interactions of the antioxidants in tissue may be modified by metabolic processes. The potential negative effects of the infusions should be included. The research performed by us is only helpful and it is not decisive about the role of the tested infusions in obstetrics. The presented preliminary studies may be continued by the use of EPR imaging to determine the localization of free radicals in tissues and free radical transformations during reactions with antioxidant plant infusions.

The relative amplitudes (A/A_DPPH_) of the EPR spectra of DPPH free radicals in contact with the individual infusions decrease to the minimum in different times (Figure 2). The values of the time (*t*_min_) to reach the minimum value by the relative amplitudes (A/A_DPPH_) of the EPR spectra of DPPH free radicals interacting with infusions of the root of *Asparagus racemosus*, *Mitchella repens* herb, *Cnicus benedictus* L. herb, *Galega officinalis* L. herb, and *Eupatorium cannabinum* L. herb not exposed to UV radiation are compared in Figure 4. The shortest times (*t*_min_), so the fastest interactions with DPPH free radicals, characterize the infusions of *Galega officinalis* L. herb and *Eupatorium cannabinum* L. herb not exposed to UV radiation. The longest times (*t*_min_) (Figure 4) indicated that the infusion of *Mitchella repens* herb interacts with DPPH free radicals the slowest.

The performed analysis of the EPR spectra of DPPH free radicals in contact with the infusions of the root of *Asparagus racemosus*, *Mitchella repens* herb, *Cnicus benedictus* L. herb, *Galega officinalis* L. herb, and *Eupatorium cannabinum* L. herb provided the information about the magnitudes and speed of free radical interactions. It was confirmed that electron paramagnetic resonance spectroscopy is helpful to study antioxidant reactions with free radicals, which is shown in this work in the example of plant infusions for obstetrics. Antioxidants, which strongly quench EPR lines of free radicals, are the best defense against free radicals.

The in vitro examined infusions of the root of *Asparagus racemosus*, *Mitchella repens* herb, *Cnicus benedictus* L. herb, *Galega officinalis* L. herb, and *Eupatorium cannabinum* L. herb., after exposition to UVA radiation, did not lose their antioxidant properties. However, it should be emphasized that their antioxidant interactions in vivo may be different from that obtained in vitro. All the infusions of the UVA irradiated raw materials, similar to infusions of nonirradiated plant materials, cause a decrease of the EPR spectra of DPPH free radicals. This effect occurs regardless of the exposure time to UVA, 30 min or 60 min. The reduction of the EPR spectra of DPPH free radicals is due to the decrease of the content of unpaired electrons in the system free radicals, the plant raw materials in the ethyl solution, resulting from the quenching of the free radicals. The quenching of free radicals is typical for antioxidant substances. The evolution of the EPR spectra of DPPH free radicals corresponds to the decrease of the EPR spectra at the first phase of the changes, and their constant position in the second phase of the changes. The infusions initially quench increasingly more free radicals, and then the situation stabilizes, and the number of free radicals remains constant in the liquid systems. The shapes of the EPR spectra of DPPH free radicals are similar because interactions with the same type of free radicals are monitored. Using one type of model of free radicals in this study allowed us to compare the antioxidant interactions of the tested infusions of different plant materials. We used a typical model of DPPH free radicals recommended in research about antioxidants [51,52,53,54,55]. Including one type of free radicals, DPPH free radicals, in all studies allowed us to compare the properties of antioxidants.

The in vitro studies using EPR spectroscopy pointed out that UV radiation influences the quenching of DPPH free radicals by the infusions of the tested plant materials. The kinetics of the free radical interactions for infusions of the root of *Asparagus racemosus* and *Mitchella repens* herb, *Cnicus benedictus* L. herb, *Galega officinalis* L. herb, and *Eupatorium cannabinum* L. herb, exposed to UV radiation for 30 and 60 min, are visualized using the time changes of the relative amplitudes (A/A_DPPH_) of EPR spectra of DPPH free radicals in Figure 5, Figure 6, Figure 7, Figure 8 and Figure 9, respectively. The kinetics depends on the type of UVA irradiated material and additionally depends on the time of exposition of the plant materials to UVA radiation (Figure 5, Figure 6, Figure 7, Figure 8 and Figure 9). In the plots in Figure 5, Figure 6, Figure 7, Figure 8 and Figure 9, the points regarding the relative amplitudes (A/A_DPPH_) of the EPR spectra of DPPH free radicals in contact with the infusions obtained from the plant materials exposed to UV radiation are located higher than the points in case of the infusions of the plant materials not exposed to UV radiation. Specifically, the quenching of free radicals by the tested infusions decreases after the exposition of the plant materials to UV radiation.

Considering the changes of the relative amplitude (A/A_DPPH_) of the EPR spectra of DPPH free radicals with increasing time of interactions (*t*) with infusions of the root of *Asparagus racemosus*, it can be noticed that the points corresponding to the data for the infusions of UVA irradiated plant material for 30 and 60 min are located above the points corresponding to the data for the infusion of the nonirradiated plant material. This situation is characteristic of a reduction of the antioxidant interactions with infusions of the root of *Asparagus racemosus* after UVA exposition. In the case of the infusion in question, this effect is rather low because the shift of the points on the graph is not large. However, some changes in the antioxidant properties argue for a rejection of these conditions of storage of the root of *Asparagus racemosus*. Specifically, the root of *Asparagus racemosus* as medicinal plant material should be protected against UVA radiation.

Analyzing the changes of the relative amplitude (A/A_DPPH_) of the EPR spectra of DPPH free radicals with increasing time of interactions (*t*) with infusions of *Mitchella repens* herb in Figure 6, we find that the points corresponding to the data for the infusions of UVA irradiated plant material for 30 min are located slightly higher than the points corresponding to the infusion of the nonirradiated material, but the points corresponding to the data for the infusions of UVA irradiated plant material for 60 min are located considerably higher. This presentation clearly points to the effect of the time of UVA irradiation of *Mitchella repens* herb on the antioxidant interactions of the infusions. UVA irradiation results in a reduction of the antioxidant properties of the infusions of *Mitchella repens* herb, and this effect is stronger for a longer irradiation time.

The changes of the relative amplitude (A/A_DPPH_) of the EPR spectra of DPPH free radicals with increasing time of interactions (*t*) with infusions of UVA irradiated *Cnicus benedictus* L. herb for irradiation times of 30 and 60 min in Figure 6 show that in this case, the interactions of the infusions of the plant material exposed to UVA are lower than for the infusion of the nonirradiated materials. The points corresponding to the infusions of the *Cnicus benedictus* L. herb exposed to irradiation times of 30 and 60 min are close together on the graphs, but they are located above the points corresponding to the infusion of the material, which was not exposed to UVA radiation. A similar situation is visible in Figure 7, which shows the changes of the relative amplitude (A/A_DPPH_) of the EPR spectra of DPPH free radicals with increasing time of interactions (*t*) with infusions of nonirradiated and UVA irradiated *Galega officinalis* L. herb. The infusions of *Galega officinalis* L. herb exposed to UVA radiation for both 30 and 60 min interact less with free radicals compared to the infusion of the *Galega officinalis* L. herb that was not exposed to radiation. Similar correlations describing antioxidant interactions were also observed for the infusions of nonirradiated and UVA irradiated *Eupatorium cannabinum* L. herb.

The values of the interactions of the tested infusion with DPPH free radicals after 12 min from UVA irradiation are compared in Figure 10. The values of the interactions correspond to the values of the relative amplitudes (A/A_DPPH_) of the EPR lines of DPPH free radicals, and the lower (A/A_DPPH_) values reflect stronger antioxidant interactions. A larger effect of UVA radiation on the free radical interactions of the tested infusions was observed for *Eupatorium cannabinum* L. herb. In this case, the (A/A_DPPH_) values shifted to values 1 after UVA irradiation.

In Figure 11, the minimal values of the relative amplitudes (A/A_DPPH_) of the EPR lines of DPPH free radicals interacting with the infusions of the root of *Asparagus racemosus* and *Mitchella repens* herb, *Cnicus benedictus* L. herb, *Galega officinalis* L. herb, and *Eupatorium cannabinum* L. herb that were not exposed to UV radiation and exposed to UVA radiation for 30 and 60 min are compared. After exposition of the plant materials to UVA radiation, minimal values of the relative amplitudes (A/A_DPPH_) of the EPR spectra of DPPH free radicals interacting with the infusions were observed, which indicates a decrease of their interactions with free radicals. During the process of increasing the exposure time of UVA radiation from 30 min to 60 min, the minimal values of the relative amplitudes (A/A_DPPH_) of the EPR lines of DPPH free radicals increased (Figure 11), which proves the decrease of the interactions of the infusions with free radicals with prolonged exposure of the plant materials to UVA radiation.

UVA radiation has the most adverse effect on the antioxidant properties of the infusion of *Eupatorium cannabinum* L. herb, for which the increase of the (A/A_DPPH_) values is higher, independent of the time of UVA irradiation (Figure 11). The weakest increase of the (A/A_DPPH_) values after UVA irradiation was observed for the infusion of the root of *Asparagus racemosus* (Figure 11). UVA radiation decreases the weakest antioxidant properties of the infusions of the root of *Asparagus racemosus*. Taking into account the deterioration of the antioxidant properties of the tested infusions after exposition of the plant materials to UVA radiation, it should be stated that the root of *Asparagus racemosus*, *Mitchella repens* herb, *Cnicus benedictus* L. herb, *Galega officinalis* L. herb, and *Eupatorium cannabinum* L. herb should not be stored under UVA radiation.

In Figure 12, the time (*t*_min_) to reach the minimum value by the relative amplitudes (A/A_DPPH_) of the EPR spectra of DPPH free radicals interacting with infusions of the root of *Asparagus racemosus*, *Mitchella repens* herb, *Cnicus benedictus* L. herb, *Galega officinalis* L. herb, and *Eupatorium cannabinum* L. herb, for infusions of the plant materials that were not exposed to UVA radiation, and exposed to UVA radiation for 30 and 60 min are compared. UVA radiation does not change the values of time (*t*_min_), and thus it does not change the speed of interactions of the infusion of the root of Asparagus racemosus, with free radicals, independent of the time of exposition to UVA radiation (Figure 12). UVA radiation reduces the time (*t*_min_); that is, it accelerates free radical interactions in case of the infusions of *Cnicus benedictus* L. herb, and *Eupatorium cannabinum* L. herb. This effect is the same for times of UVA exposure of 30 and 60 min. However, this effect is greater in the case of the infusion of *Eupatorium cannabinum* L. herb than for the infusion of *Cnicus benedictus* L. herb. In the case of the infusion of *Mitchella repens* herb, the time (*t*_min_) decreases, and the speed of interactions with free radicals increases after exposition of the plant material to UVA radiation for both 30 and 60 min. In the case of the infusion of *Galega officinalis* L. herb, the time (*t*_min_) increases after exposition of the materials to UVA radiation for 60 min, and it does not change for the exposure for 30 min. The speed of interactions of the infusion of *Galega officinalis* L. herb with free radicals decreases and does not change for the exposition to UVA radiation for 30 and 60 min, respectively (Figure 12).

Considering the values of the time (*t*_min_) to reach the minimum value by the relative amplitudes (A/A_DPPH_) of the EPR spectra of DPPH free radicals interacting with infusions of the root of *Asparagus racemosus*, *Mitchella repens* herb, *Cnicus benedictus* L. herb, *Galega officinalis* L. herb, and *Eupatorium cannabinum* L. herb., for the infusion of the nonirradiated and UVA irradiated plant materials in Figure 12, we can say that UVA radiation causes strong acceleration of free radical interactions of the infusion of *Eupatorium cannabinum* L. herb. UVA irradiation does not influence the speed of interactions of the infusion of the root of *Asparagus racemosus*, because the values of (*t*_min_) remain unchanged after exposition of this herb to UVA radiation.

Analyses of the EPR spectra of DPPH free radicals in contact with infusions of the root of *Asparagus racemosus*, *Mitchella repens* herb, *Cnicus benedictus* L. herb, *Galega officinalis* L. herb, and *Eupatorium cannabinum* L. herb showed that UVA radiation changes the magnitude and speed of free radical interactions, and these changes depend on the type of plant material and time of exposure to UVA radiation

The performed comparative examination of the infusions of the nonirradiated and UVA irradiated plant materials confirmed the usefulness of electron paramagnetic resonance spectroscopy for optimization of the storage conditions of the raw medicinal plant materials. The optimization procedure consists of detection of the changes of the EPR spectra of the plant infusions, and checking how much the spectral parameters change. The conditions under which the EPR spectra of free radicals in contact with the infusions of plant materials do not change are recommended. For the tested *Asparagus racemosus* (root), *Mitchella repens* (herb), *Cnicus benedictus* L. (herb), *Galega officinalis* L. (herb), and *Eupatorium cannabinum* L. (herb), UVA radiation is not recommended. EPR examination indicates that the tested infusions retain their antioxidant character after exposition of the plant materials to UVA radiation, but the radiation influences their interactions with free radicals.

## 3. Materials and Methods

### 3.1. Samples Preparation

The infusions of nonirradiated and exposed to UVA radiation root of *Asparagus racemosus*, *Mitchella repens* herb, *Cnicus benedictus* L. herb, *Galega officinalis* L. herb, and *Eupatorium cannabinum* L. herb were prepared. The plant materials were exposed to UVA radiation with wavelengths (λ) in the range of 315–400 nm. The UVA radiation was generated by a Medisun 250 lamp with 4 radiators with power of 20 W. The distance between the lamp and the plant materials was 30 cm. Two irradiation times were used: 30 and 60 min.

For each examined plant sample, 2 g of the herbal raw material were combined with 250 mL of boiling water and left covered for 10 min. Then, it was filtered and cooled down to room temperature.

The free radical scavenging activity of the plant infusions was examined by the use of DPPH (1,1-diphenyl-2-picryl-hydrazyl) free radicals [51,52,53,54,55]. The DPPH substance comes from Sigma-Aldrich. The control in this study was the reference solution of DPPH in ethanol. To obtain the reference solution, 5 mg DPPH in 135 mL of ethanol were mixed using a magnetic stirrer for 60 min. To examine the interactions of the tested infusions with DPPH free radicals, the individual infusions were added to the solution of DPPH in ethanol. One volume of the infusion was added to 10 volumes of DPPH solution.

### 3.2. EPR Measurements

DPPH free radicals were examined using electron paramagnetic resonance (EPR) spectroscopy. To determine the free radical scavenging activity of the tested infusions, first, the EPR spectrum of DPPH in the reference ethanol solution was measured, and then the EPR lines of DPPH free radicals in ethanol solution in contact with the infusion of the raw plant materials were obtained. For the EPR measurements, all the liquid samples were placed in thin-walled glass capillaries with a diameter of 1 mm. The capillaries were placed in the resonance cavity of the EPR spectrometer. The EPR measurements were done at room temperature. The empty capillaries did not give EPR signals. The empty capillaries did not give EPR signals at the applied receiver gains and microwave power (2.2 mW).

The first-derivative EPR spectra were measured by an X-band (9.3 GHz) electron paramagnetic resonance spectrometer of Radiopan Firm (Poznań, Poland) with magnetic modulation of 100 kHz. The numerical data acquisition system of the Rapid Scan Unit of Jagmar Firm (Kraków, Poland) was used. Microwave frequency was measured by the MCM101 recorder of EPRAD Firm (Poznań, Poland). The Nuclear Magnetic Resonance (NMR) magnetometer of EPRAD Firm (Poznań, Poland) detected the magnetic induction B of the field produced by an electromagnet. The total microwave power of the klystron in the microwave bridge of the spectrometer was 70 mW. The EPR spectra were measured with the low microwave power of 2.2 mW to avoid the microwave saturation effect in the lines. This microwave power was obtained by attenuation of 15 dB according to the formula [49]:attenuation [dB] = 10lgM/M_o_
where M_o_ and M are the total microwave power produced by klystron (70 mW) and the microwave power at which the spectrum was measured, respectively.

The exemplary EPR spectrum of DPPH free radicals is shown in Figure 13. The g-factor and the amplitudes (A) were determined for the spectra. The g-factor was calculated from the electron paramagnetic resonance condition as [48,49]:g = hν/μ_B_B_r_
where h—Planck constant, ν—microwave frequency, μ_B_—Bohr magneton, and B_r_—induction of the resonance magnetic field. The resonance magnetic induction B_r_ was determined from the EPR spectrum (Figure 13), and the microwave frequency ν was obtained by direct measurement with an NMR magnetometer.

The amplitudes (A_DPPH_) of the EPR lines of DPPH free radicals in the reference solution were compared to the amplitudes (A) of the EPR lines of DPPH free radicals in contact with the tested infusions of the plant materials. The amplitudes of the EPR spectra were calculated in arbitrary units. The relative amplitudes were determined. The amplitude of the EPR spectrum of DPPH free radicals in contact with the tested individual solution was divided by the amplitude of the EPR line of the DPPH free radicals in the reference solution (without the tested substance). The relative amplitude of the EPR spectrum of DPPH free radicals in the reference solution took a value equal 1, because it was calculated as the amplitude of DPPH in the reference solution divided by the amplitude of DPPH in the reference solution, and their values were the same. The relative amplitudes of the EPR lines of DPPH free radicals in contact with the plant infusions were lower than 1, because the infusions quenched free radicals and the amplitude of the EPR line of DPPH decreased. For the stronger antioxidants, lower values of the relative amplitude of EPR line of DPPH were observed.

As mentioned above, the free radical scavenging activity of the sample accompanied a decrease of the value of the amplitude of the EPR line of the DPPH free radicals. This effect for the known exemplary antioxidant, ascorbic acid, is illustrated in Figure 14, where the changes of the relative amplitude (A/A_DPPH_) of the EPR line of the DPPH free radicals interacting with ascorbic acid with an increasing time (*t*) of interactions are shown.

The EPR spectra of DPPH free radicals interacting with the tested plant infusions were measured for 33 min by 3 min. The changes of the amplitudes (A) of the DPPH lines with increasing interaction time were obtained. The kinetics of the interactions of the tested infusions with DPPH free radicals was determined from the changes of the amplitudes (A) of the EPR spectra of DPPH free radicals in contact with the infusions.

In the EPR studies, for the measurement of the EPR spectra of DPPH free radicals, analysis, and data presentation, the professional spectroscopic programs of Jagmar Firm (Kraków, Poland), LabVIEW 8.5 of National Instruments Firm (Austin, TX, USA), Origin (USA), and Microsoft Excel (USA) programs were used.

## 4. Conclusions

The EPR examination of free radical scavenging activity of the medicinal plant infusions obtained from the raw materials that were nonirradiated and exposed to UVA radiation pointed out that:All the tested infusions obtained from both plant materials that were nonirradiated and exposed to UVA: *Asparagus racemosus* (root), *Mitchella repens* (herb), *Cnicus benedictus* L. (herb), *Galega officinalis* L. (herb), and *Eupatorium cannabinum* L. (herb), show an antioxidant character and that they can quench the EPR spectra of DPPH free radicals.The free radical scavenging activity of the plant infusions examined in vitro depends on the type of raw plant material, and for the nonirradiated materials, it increased in the following order: *Asparagus racemosus* (root) < *Mitchella repens* (herb) < *Cnicus benedictus* L. (herb) < *Galega officinalis* L. (herb) < *Eupatorium cannabinum* L. (herb). The most effective antioxidant is the infusion of *Eupatorium cannabinum* L. (herb).For the experiment performed in vitro with the nonirradiated plant materials, the infusions of *Galega officinalis* L. herb and *Eupatorium cannabinum* L. herb interacted with free radicals the fastest. The infusion of *Mitchella repens* herb interacted with free radicals the slowest.UVA radiation changed the in vitro interactions with free radicals of all the tested infusions obtained from *Asparagus racemosus* (root), *Mitchella repens* (herb), *Cnicus benedictus* L. (herb), *Galega officinalis* L. (herb), and *Eupatorium cannabinum* L. (herb), so these plant materials should be protected from UVA radiation during storage.UVA radiation decreases the magnitudes of the interaction of the examined infusions with free radicals. This effect was most evident for the infusion of *Eupatorium cannabinum* L. herb, and it was the weakest for the infusion of the root of *Asparagus racemosus*. The in vitro free radical scavenging activity of the infusions decreased more for longer exposure times.UVA radiation caused strong acceleration in vitro of the free radical interactions of the infusion of *Eupatorium cannabinum* L. herb, but the speed of the interactions of the infusion of the root of *Asparagus racemosus* remained unchanged.EPR spectroscopy is a useful method for examining the ability of plant infusions to quench free radicals, by measurements of the changes of the amplitudes of the EPR spectra of free radicals, and these measurements in vitro are treated as preliminary research for in vivo studies. This is helpful for identifying effective antioxidants for applications in obstetrics and to determine the optimal storage conditions of medicinal herbs.

## Figures and Tables

**Figure 1 plants-10-02016-f001:**
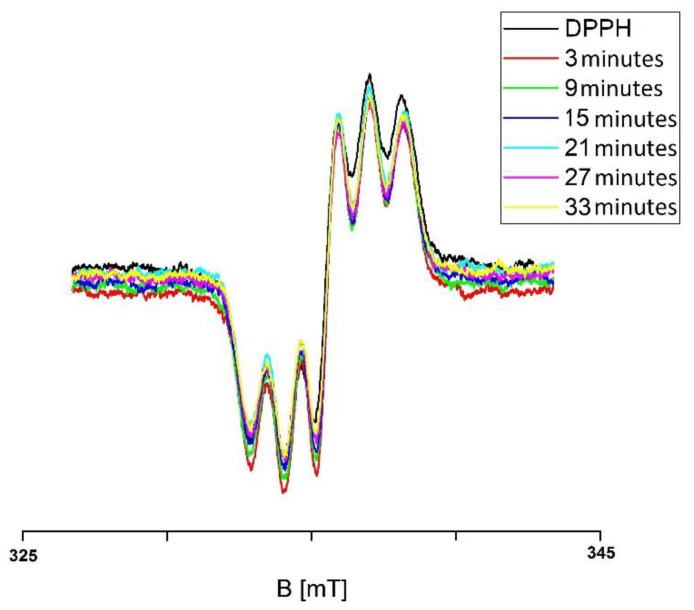
EPR spectra of DPPH free radicals interacting with the infusion of the root of *Asparagus racemosus* for 3, 9, 15, 21, 27, and 33 min. B—magnetic induction. Data for the infusion of the root of *Asparagus racemosus* not exposed to UV radiation.

**Figure 2 plants-10-02016-f002:**
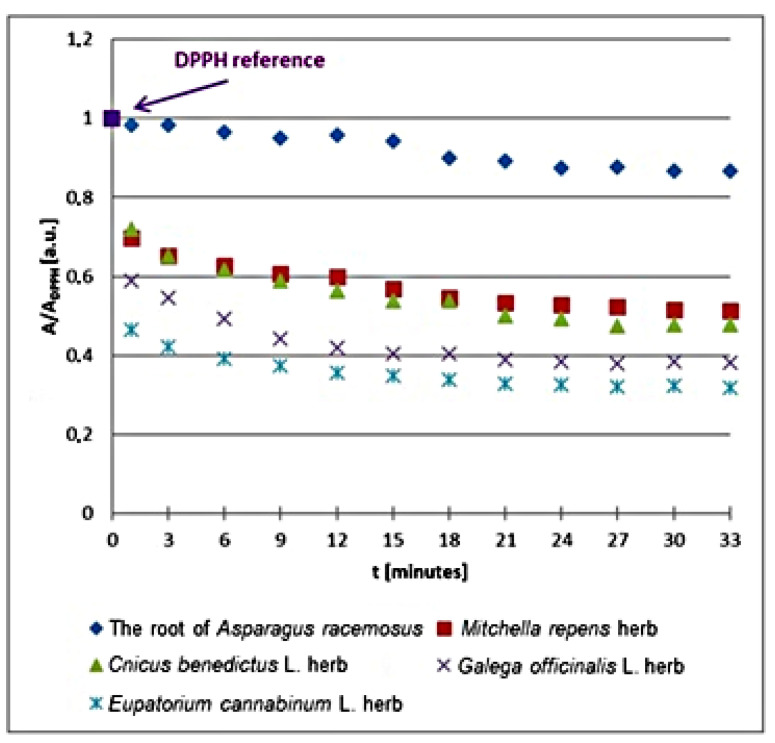
Changes of the relative amplitude (A/A_DPPH_) of EPR spectra of DPPH free radicals with increasing time of interactions (*t*) with infusions of the root of *Asparagus racemosus*, *Mitchella repens* herb, *Cnicus benedictus* L. herb, *Galega officinalis* L. herb, and *Eupatorium cannabinum* L. herb. A—amplitude of the EPR line of DPPH free radicals in contact with the tested samples. A_DPPH_—amplitude of the EPR line of DPPH free radicals in the reference solution. For the reference solution, the relative amplitude (A/A_DPPH_) takes the value of 1. Data for the infusions obtained from the plant materials not exposed to UV radiation.

**Figure 3 plants-10-02016-f003:**
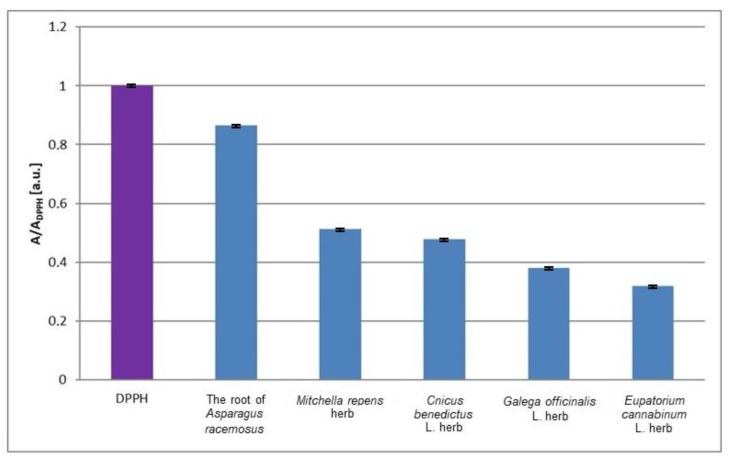
Comparison of the relative amplitudes (A/A_DPPH_) [±0.02 a. u.] of the EPR spectra of DPPH free radicals interacting with infusions of the root of *Asparagus racemosus*, *Mitchella repens* herb, *Cnicus benedictus* L. herb, *Galega officinalis* L. herb, and *Eupatorium cannabinum* L. herb for 33 min. A—amplitude of the EPR line of DPPH free radicals in contact with the tested infusion. A_DPPH_—amplitude of the EPR line of DPPH free radicals in the reference solution. For the reference solution, the relative amplitude (A/A_DPPH_) takes a value of 1. Data for the infusions obtained from the plant materials not exposed to UV radiation.

**Figure 4 plants-10-02016-f004:**
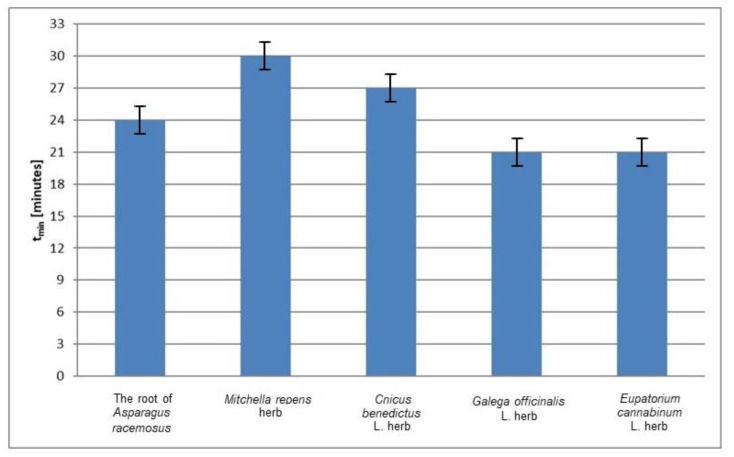
Comparison of the times (*t*_min_) to reach the minimum value by the relative amplitudes (A/A_DPPH_) of the EPR spectra of DPPH free radicals interacting with infusions of the root of *Asparagus racemosus*, *Mitchella repens* herb, *Cnicus benedictus* L. herb, *Galega officinalis* L. herb, and *Eupatorium cannabinum* L. herb. A—amplitude of the EPR line of DPPH free radicals in contact with the tested infusion. A_DPPH_—amplitude of the EPR line of DPPH free radicals in the reference solution. Data for the infusions obtained from the plant materials not exposed to UV radiation.

**Figure 5 plants-10-02016-f005:**
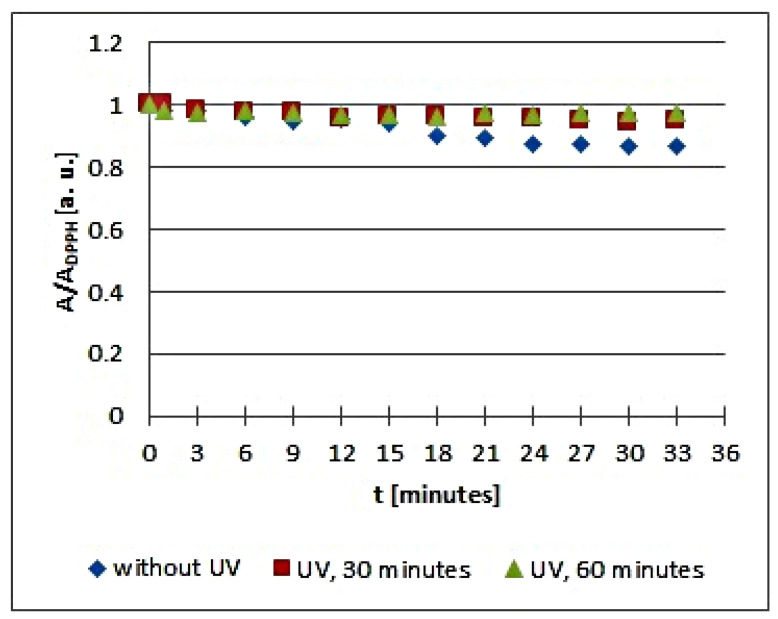
Changes of the relative amplitude (A/A_DPPH_) of EPR spectra of DPPH free radicals with increasing time of interactions (*t*) with infusions of the root of *Asparagus racemosus*. A—amplitude of the EPR line of DPPH free radicals in contact with the tested samples. A_DPPH_—amplitude of the EPR line of DPPH free radicals in the reference solution. For the reference solution, the relative amplitude (A/A_DPPH_) takes the values 1. Data for the infusions obtained from the plant materials exposed to UVA radiation for 30 and 60 min are presented. For comparison, data for the infusions of the plant material that was not exposed to UVA radiation are also given.

**Figure 6 plants-10-02016-f006:**
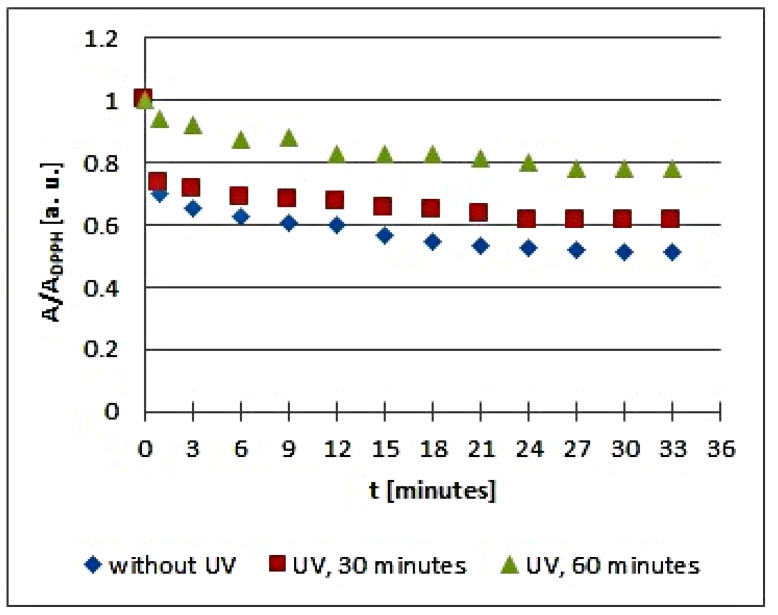
Changes of the relative amplitude (A/A_DPPH_) of EPR spectra of DPPH free radicals with increasing time of interactions (*t*) with infusions of *Mitchella repens* herb. A—amplitude of the EPR line of DPPH free radicals in contact with the tested samples. A_DPPH_—amplitude of the EPR line of DPPH free radicals in the reference solution. For the reference solution, the relative amplitude (A/A_DPPH_) takes a value of 1. Data for the infusions obtained from the plant materials exposed to UVA radiation for 30 and 60 min are presented. For comparison, data for the infusions of the plant material that was not exposed to UVA radiation are also given.

**Figure 7 plants-10-02016-f007:**
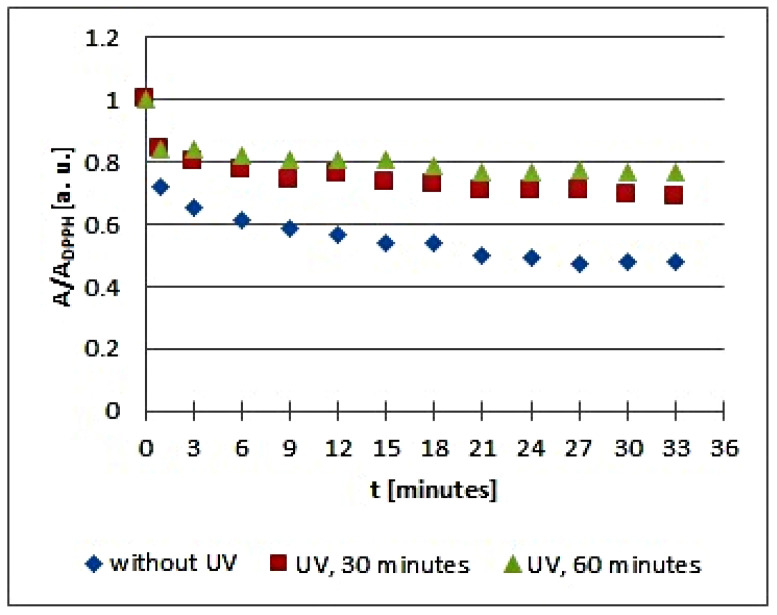
Changes of the relative amplitude (A/A_DPPH_) of EPR spectra of DPPH free radicals with increasing time of interactions (*t*) with infusions of *Cnicus benedictus* L. herb. A—amplitude of the EPR line of DPPH free radicals in contact with the tested samples. A_DPPH_—amplitude of the EPR line of DPPH free radicals in the reference solution. For the reference solution, the relative amplitude (A/A_DPPH_) takes a value of 1. Data for the infusions obtained from the plant materials exposed to UVA radiation for 30 and 60 min are presented. For comparison, data for the infusions of the plant material that was not exposed to UVA radiation are also given.

**Figure 8 plants-10-02016-f008:**
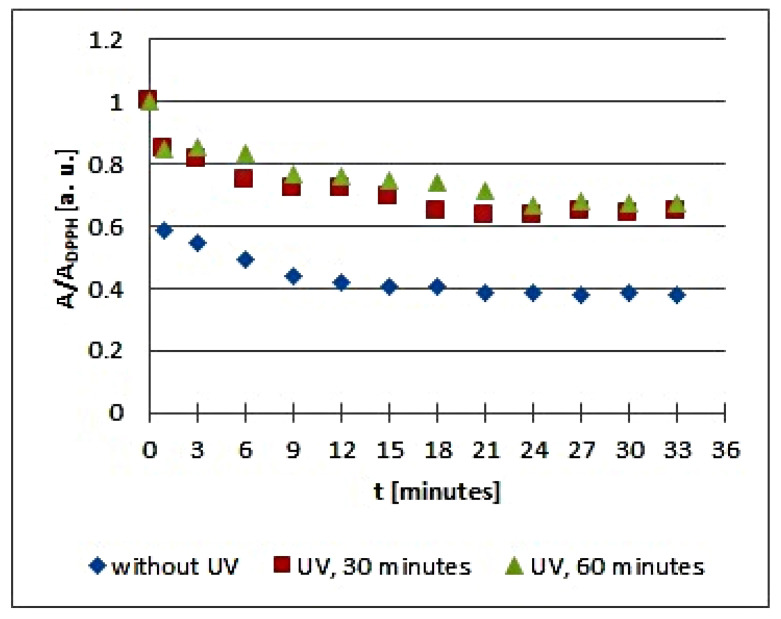
Changes of the relative amplitude (A/A_DPPH_) of EPR spectra of DPPH free radicals with increasing time of interactions (*t*) with infusions of *Galega officinalis* L. herb. A—amplitude of the EPR line of DPPH free radicals in contact with the tested samples. A_DPPH_—amplitude of the EPR line of DPPH free radicals in the reference solution. For the reference solution, the relative amplitude (A/A_DPPH_) takes a value of 1. Data for the infusions obtained from the plant materials exposed to UVA radiation for 30 and 60 min are presented. For comparison, data for the infusions of the plant material that was not exposed to UVA radiation are also given.

**Figure 9 plants-10-02016-f009:**
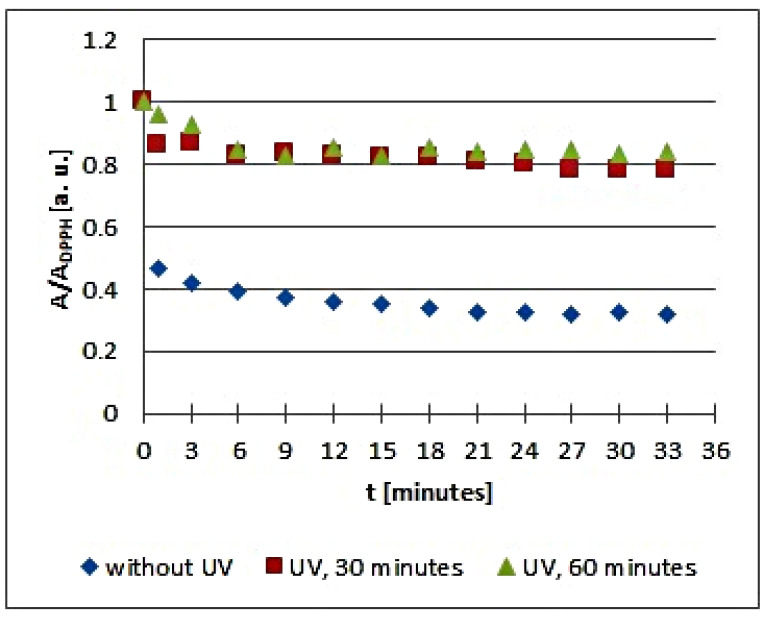
Changes of the relative amplitude (A/A_DPPH_) of EPR spectra of DPPH free radicals with increasing time of interactions (*t*) with infusions of *Eupatorium cannabinum* L. herb. A—amplitude of the EPR line of DPPH free radicals in contact with the tested samples. A_DPPH_—amplitude of the EPR line of DPPH free radicals in the reference solution. For the reference solution, the relative amplitude (A/A_DPPH_) takes a value of 1. Data for the infusions obtained from the plant materials exposed to UVA radiation for 30 and 60 min are presented. For comparison, data for the infusions of the plant material that was not exposed to UVA radiation are also given.

**Figure 10 plants-10-02016-f010:**
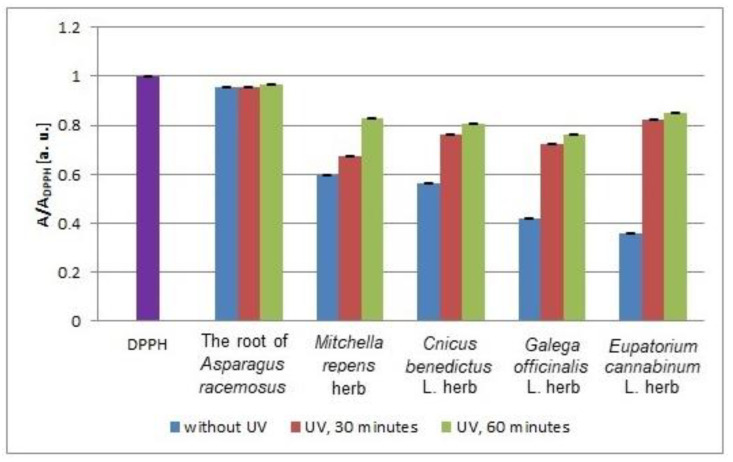
Comparison of the relative amplitudes (A/A_DPPH_) [±0.02 a. u.] of the EPR spectra of DPPH free radicals interacting with the infusions of the root of *Asparagus racemosus* and *Mitchella repens* herb, *Cnicus benedictus* L. herb, *Galega officinalis* L. herb, and *Eupatorium cannabinum* L. herb for 12 min. A—amplitude of the EPR line of DPPH free radicals in contact with the tested samples. A_DPPH_—amplitude of the EPR line of DPPH free radicals in the reference solution. For the reference solution, the relative amplitude (A/A_DPPH_) takes a value of 1. Data for the infusions obtained from the plant materials exposed to UVA radiation for 30 and 60 min. Data for the infusions of the plant material that was not exposed to UVA radiation are also given.

**Figure 11 plants-10-02016-f011:**
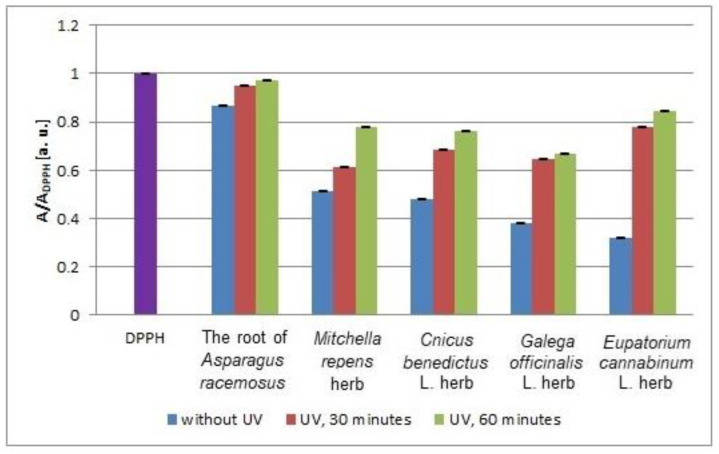
Comparison of the relative amplitudes (A/A_DPPH_) [±0.02 a. u.] of the EPR spectra of DPPH free radicals interacting with infusions of the root of *Asparagus racemosus* and *Mitchella repens* herb, *Cnicus benedictus* L. herb, *Galega officinalis* L. herb, and *Eupatorium cannabinum* L. herb for 33 min. A—amplitude of the EPR line of DPPH free radicals in contact with the tested samples. A_DPPH_—amplitude of the EPR line of DPPH free radicals in the reference solution. For the reference solution, the relative amplitude (A/A_DPPH_) takes a value of 1. Data for the infusions obtained from the plant materials exposed to UVA radiation for 30 and 60 min. Data for the infusions of the plant material that was not exposed to UVA radiation are also given.

**Figure 12 plants-10-02016-f012:**
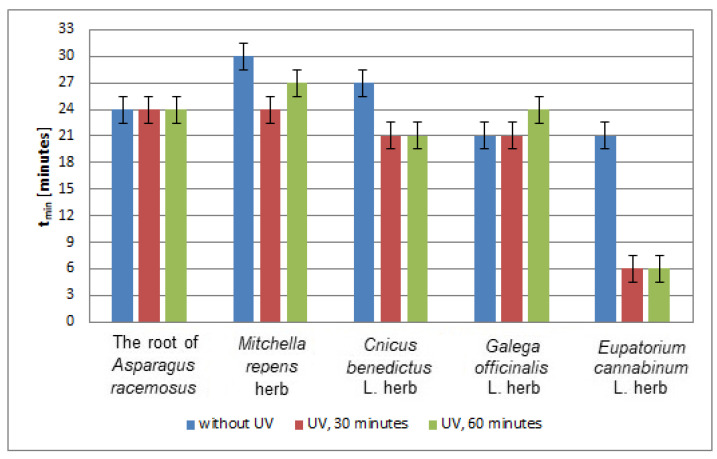
Comparison of the times (*t*_min_) to reach the minimum value by the relative amplitudes (A/A_DPPH_) of the EPR spectra of DPPH free radicals interacting with infusions of the root of *Asparagus racemosus*, *Mitchella repens* herb, *Cnicus benedictus* L. herb, *Galega officinalis* L. herb, and *Eupatorium cannabinum* L. herb. A—amplitude of the EPR line of DPPH free radicals in contact with the tested samples. A_DPPH_—amplitude of the EPR line of DPPH free radicals in the reference solution. Data for the infusions obtained from the plant materials exposed to UVA radiation for 30 and 60 min. Data for the infusions of the plant material that was not exposed to UVA radiation are also given.

**Figure 13 plants-10-02016-f013:**
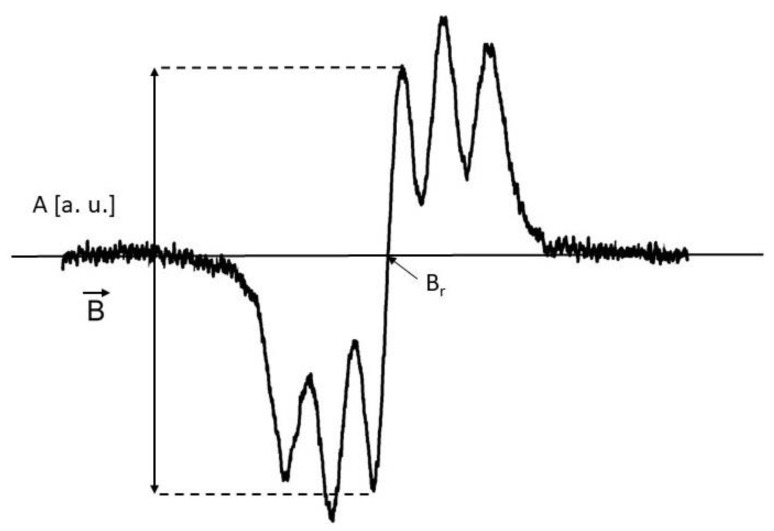
The exemplary EPR spectrum of DPPH free radicals in ethyl alcohol solution. A—amplitude of the EPR lines, B—magnetic induction, B_r_—resonance magnetic induction.

**Figure 14 plants-10-02016-f014:**
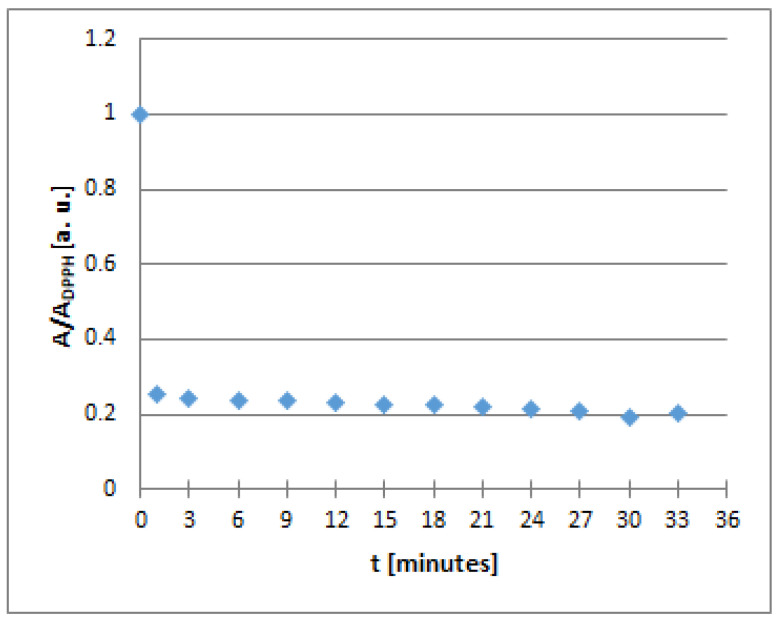
Changes of the relative amplitude (A/A_DPPH_) of EPR spectra of DPPH free radicals with increasing time of interactions (*t*) with ascorbic acid. A—amplitude of the EPR line of DPPH free radicals in contact with ascorbic acid. A_DPPH_—amplitude of the EPR line of DPPH free radicals in the reference solution. For the reference solution, the relative amplitude (A/A_DPPH_) take a value of 1.

## Data Availability

Data available on request from authors. The data that support the findings of this study are available from the corresponding author upon reasonable request.

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
