# Peer review of "Free Radical Scavenging Activity of Infusions of Different Medicinal Plants for Use in Obstetrics"

_plants, 2021, doi:10.3390/plants10102016_

Round 1
Reviewer 1 Report
The authors have done a good job following the reviewers' suggestions thus, the revised version of the manuscript is improved.
I have just one comment about the number of self-citations that I think, is too much (17 is quite a lot!). I suggest the authors reduce them.
Author Response
Professor of physics Barbara Pilawa 6 September, 2021
Head of Department of Biophysics
Faculty of Pharmaceutical Science in Sosnowiec
Medical University of Silesia in Katowice
Jedności 8, 41-200 Sosnowiec, Poland
e-mail: bpilawa@sum.edu.pl
Editor-in-Chief of The Plants Journal
Professor Dilantha Fernando
I am sending the corrected manuscript of paper entitled Free radical scavenging activity of infusions of different medicinal plants for use in obstetrics prepared by Sylwia Jarco, Barbara Pilawa, and Paweł Ramos.
We appreciate the valuable comments on our research. All the valuable comments of the Reviewers have been taken into account. According the Reviewer 1 we reduced self-citations form 17 to 2 (citations numbers: 46 and 47). The introduction was re-written according to the remarks of the Reviewer 3. As was suggested by the Reviewer 3 Results and Discussion were combined into one paragraph and the duplicated information were removed. The Figures indicated by the Reviewer 3 were removed. The editorial errors have been corrected. The EPR examination of interactions of the ascorbic acid with DPPH free radicals was performed. As was suggested the kinetics of interactions of DPPH with ascorbic acid as the changes of amplitudes of EPR lines of DPPH was shown. The results were introduced to the corrected paper.
I will be grateful if You find the corrected paper suitable for publication in Plants.
Sincerely Yours
Barbara Pilawa

Reviewer 2 Report
In the presented article, the authors discussed the topic of antioxidant activity of infusions from various medicinal plants in obstetrics.
Work thought out and well structured. The manuscript has all the necessary elements, the goal is clearly stated, the results are clearly visualized and described in detail, and the whole is summarized with conclusions.
An interesting topic and the research methods used will surely interest the recipients.
In my opinion, this article deserves a publication in Plants.
Author Response

(The authors gave the same response as above.)

Reviewer 3 Report
Introduction should be strongly re-written. Authors should focused to provide relevant background and they should avoid to write about their investigation. The aim of the study should be clearly described in one paragraph. In lines 43-50 there are unnecessary repetition. Lines 58-83: such detailed description is unnecessary . Line 135-139 – another repetition regarding on Authors experiments
Line 119: „because it accumulates the secretion of prolactin” - increases the secretion of prolactin?
Line 160: „(…) are shown below…” – are shown on Fig….
Results should be presented in more compact form. Too many figures – most of them should be moved to Supplementary material (Fig.1-5, 9-10,12-13, 15-16, 18-19, 21-22. One/two representative examples to show EPR spectra are enough.
Changes of the relative amplitude are more informative.
Figs. 27-28 are unnecessary. They are too elementary
The quality of some Figures is poor.
Discussion is rather description of Authors results. A lot of information are duplicated in both Result and Discussion sections. Maybe combining of both paragraphs into one will be better?
A known antioxidant (e.g. ascorbic acid) should be included in the investigation to prove that the changes of amplitude are due to the antioxidant property.
There are some editorial errors, e.g. lack of italic (line 87, 89…, Figure 24-26).
Author Response

(The authors gave the same response as above.)

Round 2
Reviewer 3 Report
Manuscript has been corrected and I recommend it for publication in current form